# Feasibility Study on Hyperspectral LiDAR for Ancient Huizhou-Style Architecture Preservation

**Hui Shao** [1,2,3] , **Yuwei Chen** [2,4,*] , **Zhirong Yang** [5] , **Changhui Jiang** [2,6] , **Wei Li** [4] , **Haohao Wu** [4] , **Shaowei Wang** [7] , **Fan Yang** [7] , **Jie Chen** [1,3] , **Eetu Puttonen** [2] and **Juha Hyyppä** [2]

1   The School of Electronic and Information Engineering, Anhui Jianzhu University, Hefei 230601, China; shaohui@ahjzu.edu.cn (H.S.); chenjie@ahjzu.edu.cn (J.C.)
2   Center of Excellence of Laser Scanning Research, Finnish Geospatial Research Institute, FI-02430 Masala, Finland; changhui.jiang1992@gmail.com (C.J.); eetu.puttonen@nls.fi (E.P.); juha.hyyppa@nls.fi (J.H.)
3   Key Laboratory of Huizhou Architecture in Anhui Province, Hefei 230601, China
4   Key Laboratory of Quantitative Remote Sensing Information Technology, Chinese Academy of Sciences, Beijing 100094, China; liwei@aoe.ac.cn (W.L.); hhwu@aoe.ac.cn (H.W.)
5   Department of Computer Science, Norwegian University of Science and Technology, NO-7491 Trondheim, Norway; zhirong.yang@ntnu.no
6   Interdisciplinary Division of Aeronautical and Aviation Engineering, The Hong Kong Polytechnic University, Kowloon, Hong Kong, China
7   Shanghai Institute of Technical Physics, Chinese Academy of Sciences, Shanghai 200083, China; wangshw@mail.sitp.ac.cn (S.W.); fanyang@mail.sitp.ac.cn (F.Y.)
*   Correspondence: yuwei.chen@nls.fi

**Abstract:** Huizhou-style ancient architecture was one of the most important genres of architectural heritage in China. The architecture employed bricks, woods, and stones as raw materials, and timber frames were significant structures. Due to the drawback that the timbers were vulnerable to moisture and atmospheric agents, ancient timber buildings needed frequent protective interventions to maintain its good condition. Such interventions unavoidably disrupted the consistency between the original timber components. Besides this, the modifications brought about difficulty in correctly analysing and judging the state of existing ancient buildings, which, in current preservation practices, mainly rely on the expertise of skilled craftsmen to classify wood species and to identify the building-age of the timber components. Therefore, the industry and the research community urgently need a technique to rapidly and accurately classify wood materials and to discriminate building-age. In the paper, we designed an eye-safe 81-channel hyperspectral LiDAR (HSL) to tackle these issues. The HSL used an acousto-optic tunable filter (AOTF) as a spectral bandpass filter, offering the HSL measurements with 5 nm spectral resolution. Based on the HSL measurements, we analysed the relationship between the surface and cross-section spectral profiles of timber components from different ancient architectures built in the early Qing dynasty (~300 years), late Qing dynasty (~100 years), and nowadays, and confirmed the feasibility of using surface spectra of timber components for classification purpose. We classified building-ages and wood species with multiple Naive Bayes (NB) and support vector machine (SVM) classifiers by the surface spectra of timber components; this also unveiled the possibility of classifying gnawed timber components from its spectra for the first time. The encouraging experimental results supported that the AOTF-HSL is feasible for historic timber building preservation.

**Keywords:** Huizhou-style; ancient architecture; hyperspectral LiDAR; classification

## 1. Introduction

Huizhou-style ancient architecture was one of the most important genres of architectural heritage in China. It extensively employed timber components, especially the load-bearing skeletons and the main architectural frames with rational layouts and sophisticated decorations. The architecture reflected mountainous features of the Eastern part of China and the harmony between traditional Chinese religions and their living style. Such historic timber buildings were witnesses to a rich tradition of craftsmanship, cultural values, and structural and material knowledge (Figure 1a). Due to its significance in human history, in 2000, the United Nations Education, Scientific and Cultural Organization (UNESCO) chose it as world heritage.

These relics of the old way of life and the evidence of elaborate construction skills inherited from the past are gradually disappearing. The structural problems in historic timber buildings are caused by various factors, for example, destructive natural hazards, imprudent man-made alterations, or chronic lack of maintenance, eventually leading to major structural damage [1]. Most Huizhou-style ancient architectures are located in the areas with subtropical monsoon climates (Figure 1b) and such humid climates introduce natural threats to the timber structure, such as threats of hurricane and flood, and moisture in the rain season; besides insect damage, man-made unwise modifications and fire all impose threats to the timber architectures.

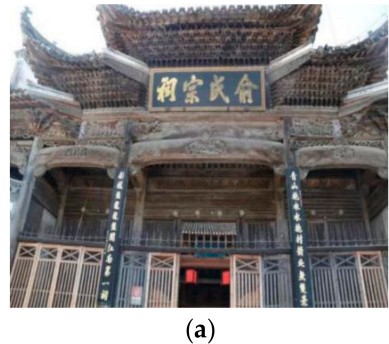
(**a**)

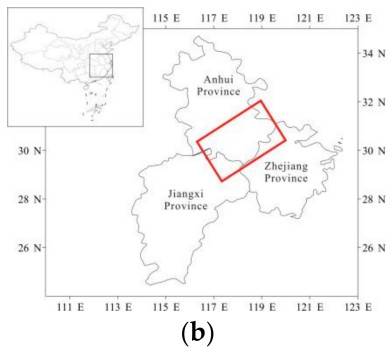
(**b**)

**Figure 1.** (**a**) The typical Huizhou-style ancient architecture, (**b**) the sketch map of China showing the distribution area of ancient Huizhou-style architectures (in red rectangle).

To prevent the decay of the ancient timber buildings, various conservation approaches, such as regular maintenance or constant component replacement, are necessary to keep the heritage buildings in good condition, and also to preserve the cultural values embedded in them. According to the latest released from Relics Conservation Law of the People's Republic of China [2], protection strategies, approaches, and implements must strictly follow the rule 'no originality to be changed'. Additionally, 'Originality' was defined as the combination of architectural and structural patterns, fabrics, and construction skills [3]. Previously, conservation practices appreciated a 'fresh' appearance more as the perception of the achievement of the repair work [4]. Nowadays, such perception is challenged by the appreciation of authenticity; preserving or freezing an ancient building in its present state is more encouraged in current conservation practices.

The methodology of conservation regarding heritage timber buildings mainly includes four aspects [5]: (1) Comprehending the evolution of their behaviour in the temporal dimension; (2) formulating a diagnosis relating to their current state in the spatial dimension; (3) forecasting their future performance; and (4) devising appropriate measures of intervention. Thus, understanding the current status of ancient timber buildings in the spatial dimension and comprehending its evolution in history are the premises for carrying out protective intervention for better future performance. Recording the state of ancient complex buildings precisely is the basis of heritage protection; we desire to use noninvasive instruments with the minimum number of sensors, which follows the international heritage buildings preservation criteria and protocols [6], and desire to distinguish structural changes,

materials, and others information regarding the structures [7]. However, identifying temporal knowledge and classifying material species are often left out of such preservation applications.

Figure 2a shows that the ancient timber temple had been historically repaired three times in the Qing Dynasty (1636–1912), according to the repaired documents since it was originally built from 1506–1521 in the Ming Dynasty. We can observe that the repaired and replaced parts followed the original design, while the appearance of the replacements and original parts were too similar to be discriminated by visual inspection, even with a skilled craftsman or traditional equipment. Figure 2b shows a reconstruction work conducted recently. While materials and styles follow the original state in the reconstruction process, there are many materials and crafting's of components that differ from the original work, that is, the new replacement has a different appearance (patina) with the remaining fabrics, and the patina is a film on the wood surface, produced by oxidation over a long period or sheen produced by age. From the images, we can conclude that the historical and present modifications might bring extra difficulty in correctly analysing and judging the 'original state' and 'repaired state' from existing ancient buildings, which, in current preservation practices, mainly rely on the empirical judgment of skilled craftsmen to classify wood species and to discriminate the building-age of timber components.

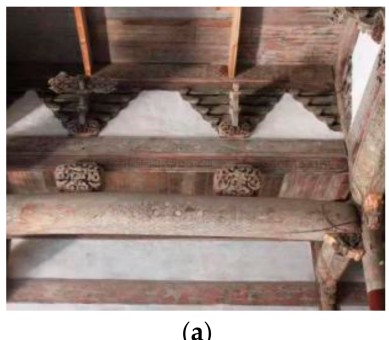
(a)

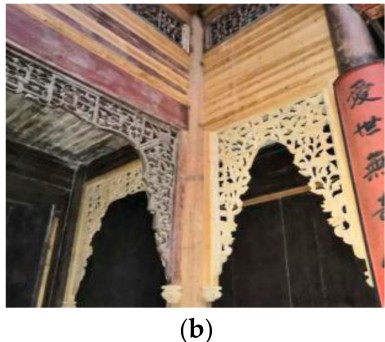
(b)

**Figure 2.** The repaired Huizhou-style buildings of timber components in the (**a**) JiXi Confucian Temple (which had been historically repaired and restored three times during Qing dynasty) and (**b**) Chengzhi Tang in Hong Village HuangShan, Anhui Province.

Historical records for the renovations are essential for the buildings state assessment, but it is common that records were ambiguous or missing for official buildings, such as ancestral temples and temple architecture; there was not even a record for private residential houses. It definitely multiplies the difficulties in forecasting its future performance and devising appropriate intervention. Therefore, it is necessary to employ an immediate survey and measurements for the state assessment and material judgement of the present ancient timber buildings.

To record and survey historic timber buildings, there are four different solutions adopted, including manual measurements [8,9], topographic methods [10,11], close-range photogrammetry [12], and laser scanning methods [13,14]. Due to their lower efficiency and precision, traditional manual measurements have gradually been replaced by instrumental documentation [9]. The topographic methods provide an essential support in assessing the structural analysis of ancient buildings, which surveys and monitors the anomalies and deformations of buildings for a conscious conservation [15]. However, they are not suited for the complex, irregular geometry structures, and need large instrumental efforts to obtain a large number of measurement points [16]. Close-range photogrammetry and laser scanning technologies can obtain the geometry of the superficial contour of ancient buildings, even in the case of irregular geometric shapes [16]. They can obtain excellent solutions to map and record ancient buildings, and can also provide a noncontact solution for data acquisition [17]. Laser scanning methods provide several unique advantages with respect to other methods, for instance, enabling the analysis of inaccessible structural elements, obtaining 3D surface information without direct contact with samples, efficiently collecting data in the form of ready-to-use 3D point clouds with high spatial resolution,

and analyzing the texture of the sample's surface, independent of illustration conditions [18]. Most laser scanners, employed widely in 3D reconstruction, operate at a single-wavelength, providing high spatial resolution point data with intensity information on a selected wavelength. However, they are insufficient in obtaining different characteristics according to the scattering properties of the building components [19]. Furthermore, the intensity information derived from a single-wavelength LiDAR is not optimal for wood classification and building-age discrimination, especially for the delicate appearance difference of timber components. Various factors may mislead the classification results, for example, incident angle, wood texture, and patina of wood, and thus temporal evolution evaluation becomes an impossible mission for the single-wavelength LiDAR technique.

To assess the condition of heritage timber structures, we need a deep understanding of their past and current states [20]. The best solution for conservation and protection is founded on the combination of an adequate understanding, comprehensive analysis, and risk assessment. First of all, such a combination can be fulfilled when the following issues are resolved: (1) Classifying the wood species of timber components; (2) discriminating components with different building-age to recognize the trace of renovation; and (3) separating damaged parts from the undamaged frames, such as gnawed timber, which is one of the most common and serious threats to the ancient buildings in the Huizhou area [21,22]. In heritage preservation, the widely used laser scanners can obtain dense 3D points on the object's surface with high accuracy in order to record geometric and spatial knowledge of ancient buildings [23,24]. While we trace the historical pattern and gradual evolution in construction practice, such laser scanners are not competent in solving these problems. A new technique, enabling the recording of the detailed spatial and spectral information, will be of great interest in both the research community and the organizations for ancient timber architecture protection. Therefore, LiDAR scanners that can operate with more than one spectral wavelength are highly anticipated.

The development of active hyperspectral LiDAR (HSL) becomes possible due to the availability of commercial supercontinuum lasers [25,26]. Over the past decades, the utility of HSL in target classification has been a topic of interest [26–30]. HSL is a non-contact method that minimizes the measurement time and allows us to obtain the section properties, which can be combined together with some material testing characterization in order to obtain highly automatic data processing [27]. Spectra, acquired with active hyperspectral instruments, are not affected by illumination conditions or even shadows, thus significantly simplifying post-processing [28]. Additionally, the HSL was proven to classify similar materials effectively and to extract vegetation parameters based on the collected spectral information [29–32]. The classification of timber components based on the HSL measurement will help to estimate 'origin state', which may provide a new tool to study and research historical evolution in the heritage architecture field.

In the paper, we designed an eye-safe tunable HSL based on a supercontinuum laser and acousto-optic tunable filter (AOTF) device, termed AOTF-HSL in the following context. We evaluated the feasibility of employing surface spectra for classification by comparing it with the cross-section spectra derived from the AOTF-HSL, and then explored it in order to discriminate the building-ages and to classify the wood species of the timber components by AOTF-HSL measurements with multiple Naive Bayes (NB) and support vector machine (SVM) classifiers. The investigation was conducted by discriminating the building-age difference of timber components in order to distinguish the 'original components' from 'repaired components', and to further classify wood species from the same ancient architecture, which have similar appearances due to the long-term exposure to an open-air circumstance. The classification of the gnawed sample and ancient timber samples was also performed in a laboratory circumstance.

The major contributions of this research are listed as follows:

1. We confirmed the consistency of spectra collected by the AOTF-HSL from both the surface and the cross-section parts of the timber samples.
2. We classified wood species and discriminated the building-age of the timber components from ancient architectures based on the AOTF-HSL derived spectral information.

3.  For the first time, we revealed the possibility of employing the AOTF-HSL measurements to classify the gnawed timber component.

The remainder of the paper is organized as follows: Section 2 introduces the details of the HSL system based on an AOTF. Section 3 outlines samples and classification methods. Section 4 presents results and an analysis of the laboratory experiments on timber samples classification. Finally, in Section 5, the conclusions are drawn.

## 2. Instrument and Measurements

Most of the developed HSLs have the characteristic of high-energy pulse, which is not an eye-safe solution and might cause damage on the surface of the detected timber components. The developed HSL also has limited or discrete spectral channel system configuration [33,34]. To tackle this problem, we designed and tested a tunable HSL technique, based on an AOTF device [29,30]. In this paper, the AOTF-HSL with the finer spectral resolution (5 nm) was developed to evaluate the surface spectrum of the various timber component samples taken from ancient Huizhou-style architectures.

### 2.1. AOTF-HSL

Figure 3 illustrates the current AOTF-HSL setup. Owing to a supercontinuum laser source with an ultra-wideband spectral range from 450 to 2400 nm, the AOTF-HSL has fine spectral resolution and consecutive bands to satisfy the demands of the field test of consecutive and high spectral resolution profiles. The AOTF modulates the intensity and wavelength of a laser beam simultaneously, and we selected its spectrum with coverage from 650 to 1050 nm. A Cassegrain telescope with 100 mm aperture in diameter and 700 mm focal length was employed as a receiving optic to collect the scattered laser pulse from the targets. The avalanche photodiode (APD) sensor converted the laser echo into an electronic signal and then amplified it. The major parameter specifications of AOTF-HSL are listed in Table 1. The AOTF-HSL output 81 channels of waveform echoes covering 650–1050 nm with a 5 nm spectral resolution for the sampling points was the target in this research.

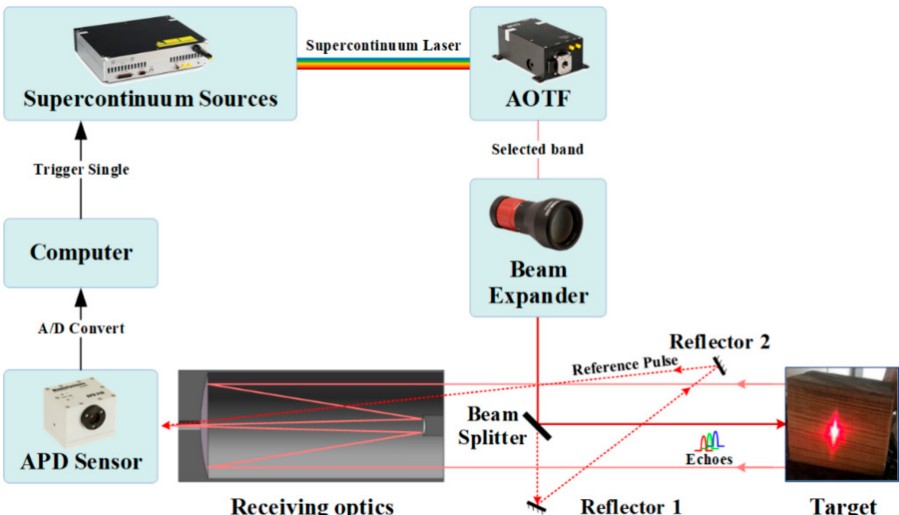

**Figure 3.** The schematic acousto-optic tunable filter–hyperspectral LiDAR (AOTF-HSL) setup illustration.

**Table 1.** The AOTF-HSL major parameter specifications.

| Parameter | Description/Value |
|---|---|
| Spectral range | 650–1050 nm |
| Spectral resolution | 2–10 nm |
| Beam divergence | 0.4 mill radian |
| Beam diameter (at exit) | 10 mm |

The data collecting experiments were conducted under a controlled laboratory environment (Figure 4) to obtain range measurements and hyperspectral information simultaneously from the waveforms collected via the AOTF-HSL. Since the range performance and the accuracy of the collected spectrum have been evaluated in previous researches [26–29], in this paper, such issues are not discussed. For the 0.4 mill radian beam divergence, the AOTF-HSL had a small focusing spot size to acquire more refined detail information and the texture of ancient buildings. We mainly focused on the feasibility study as to how to utilize the HSL measurements to discriminate the building-age of ancient timber components and to classify their wood species.

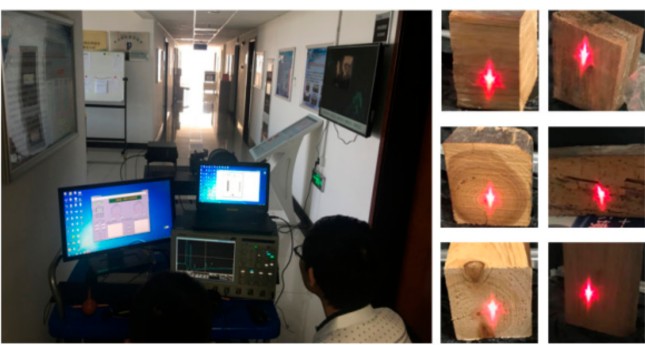

**Figure 4.** The data collecting experiment and some timber samples under testing.

### 2.2. Distance Measurements

The distance between the targets and the AOTF-HSL was approximately 18 m to collect the backscattered signals by a high-speed oscilloscope with a 20 G samplings/sampling rate. The small laser spot is more easily reflected in the sample surface, which is diffuse with high reflectivity, resulting in a better reflection of the signal back to the scanner [35]. Meanwhile, the selected distance of tens of meters offers a feasible effective radius for a practical survey and record it in future with an in-site terrestrial scanning mode. The time-of-flight (ToF) measurements of different spectral channels are calculated based on the collected waveforms maximum values [26].

### 2.3. Hyperspectral Measurements

The echo intensity is normalized with the intensity of the echo collected from a reference panel at the same measurement distance, converting into backscattered reflectance [27]. The reflectance spectra can be employed to produce a hyperspectral point cloud, combined with the corresponding ToF and concurrent scanner orientation. Before collecting the reflectance spectra of timber samples, a $10 \times 10$ cm white reference panel with 70% reflectivity (Spectralon, Labsphere Inc.) is placed at an identical distance as the samples away from the HSL system; spectral profiles were captured first, and then the capture was performed repeatedly to weaken the influence of some external factors.

To ensure that the intensities are internally comparable, we employ the backscattered reflectance (R) that is converted from radiances ($R_i$) by normalizing from the panel radiances ($R_{ref}$) measured at the same channel:

$$R(\lambda) = R_i(\lambda)/R_{ref}(\lambda), \tag{1}$$

The panel is not an authentic Lambertian object in the backscatter direction, and the calculated reflectance does not strictly accord with the definition for reflectance factor [36]. However, it will provide a calibrated reflectance to simplify the spectra, which is based on the transmitted spectrum and the target distance. The property of the target collected by the HSL can represent physical timber component characteristics well. However, the measurement process may be affected by various factors, such as the surface oxidization and human vandalism, etc., and hence data pre-processing is necessary, which is demonstrated to improve the accuracy of classification [37].

## 3. Samples and Methods

### 3.1. Samples

To evaluate the performance of the AOTF-HSL in ancient architecture preservation, nine timber samples, with four wood species, were employed in laboratory experiments, respectively, pine (*Pinus sylvestris* L.), spruce (*Picea abies* (L.) H. Karst.), papyrifera (*Broussonetiapapyrifera* (I)vent), and hawthorn (*Crataegus pinnatifida* Bunge), as listed in Table 2.

**Table 2.** The samples of timber components.

| Species | 0-Year | 100-Year | 300-Year |
|---|---|---|---|
| Pine | √ | √ | √ |
| Spruce | √ | √ | √ |
| Hawthorn | | √ | |
| Papyrifera | | √ | |
| Gnawed pine | | √ | |

We divided the samples into three categories: The first one was the samples freshly cut two months from the forests used as a partial replacement of the damaged parts, which were cut and scraped, and then they were kept outside to make sure that they were suitable for dehydration, we termed them as 0-year samples in the following context. The second and third categories: Indoor ancient building component samples, which were sawed off from the timber components of ancient buildings during the renovation, were termed according to the building ages of 100-years and 300-years. The surface of samples were decayed and darkened due to the natural oxidization caused by long-term exposure to air.

### 3.2. Classification Methods

The wood species and building-age of timber components are classified using two methods: Naive Bayes and support vector machines.

Naive Bayes or the NB classifier is a straightforward and frequently used method for supervised learning. NB applies the Bayes theorem with independent assumptions [38–40]. The NB classifier can easily and quickly be implemented with a small number of parameters. Therefore, little storage space is required during training and classification. We can find all the probability instances and attributes required in one scan and update the model easily. Moreover, the NB classifier takes into account evidence from many attributes to make the final prediction very transparent. In this way, it is easy to find that probabilistic explanations replicate their way of diagnosing. As a result, we can pick the important channels out of the supercontinuum channels to further simplify the hardware.

A support vector machines or SVM classifier is another supervised learning method, which is built on the statistical learning theory presented by Vapnick [41] and is based on structural risk minimization. SVM is less sensitive to training sample size. Improved SVMs have been proposed to work with limited training samples. SVMs seek a hyperplane that maximizes the margin between classes and uses a few boundary points called support vectors to create the decision boundary [42]. SVM can be equipped with different kernels to handle various nonlinear decision boundaries. We chose the Gaussian Radial Basis Function or RBF kernel because it is stationary, isotropic, and smooth. RBF is

one of the most popular kernels used in SVM [42,43], which works well in practice and is relatively easy to tune. Given a precision s > 0, the RBF kernel is defined as

$$K(x_i, x_j) = \exp(-s \|x_i - x_j\|^2),$$

We have used Python and the Scikit-learn software package, which includes the SVM classifier in this research [44]. Each species listed in Table 2 was scanned using the AOTF-HSL using the AOTF-HSL eight times, and each scanning could generate 81-channel spectral intensities with 4000 reflectance values. The collected spectral values were employed to determine the maximum intensity value. To evaluate the classification performance, we followed the standard machine learning process to divide the data set: In each randomization, four measurements of each sample were assigned to generate the training dataset, and the remaining four measurements were used to generate the testing dataset.

## 4. Experiments and Analysis

### 4.1. Feasibility Test of Surface Spectrum

The timber frame structures are vulnerable to atmospheric agents, such as light, water, and smoke, in spite of the indoor environment. The varying degree of surface timber components increased with age, which has yet to be fully confirmed, the different species with similar surfaces are not clearly discernible by the backscattered intensities of the laser. Therefore, the property of the laser intensity is greatly affected by the timber surface characteristics. To verify whether the 81-channel AOTF-HSL measurements of the surface of ancient timber components can be employed to discriminate building-age and to classify wood species, we first attempted to find the relationship between the surface and cross-section measurements.

The measurements of the cross-section may be reliable for the lower ageing speed of the interior of the timber component; we newly sawed samples from timber structures for testing, however, we could not obtain spectra of the cross-section in practical application. The 81-channel measurements of the surface and cross-section of the same component are collected, as Figure 5 shows.

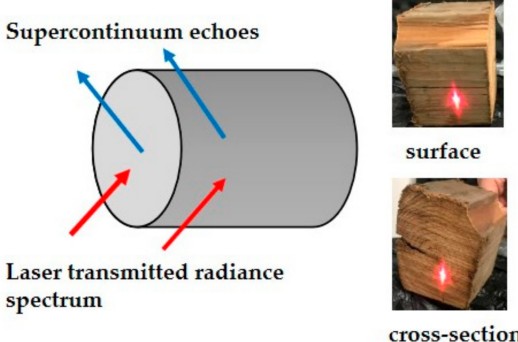

**Figure 5.** Experiment schematic diagram of surface and cross-section measurements.

Figures 6 and 7 present the surface and cross-section echo signal maximum and reflectance distribution tendency of pine and spruce samples with different building-ages. As can be seen, the tendency of surface and cross-section reflectance distribution is consistent, not only in pine samples, but also spruce samples. In particular, the reflectance value tendency of spruce and pine samples, based on building-age, is adverse, the maximum echo values and reflectance of pine increase with building-age generally, as Figure 6 shows, while the maximum echo values and reflectance of spruce decreases with building-age, as shown in Figure 7. The results coincide with the judgments of the building-age of ancient timber architectures by the expertise of craftsman. Such judgments are based on the timber's patina features: When oil-rich pine timbers are employed as building components, they would present glossy appearances for the gradually de-oiling procedure, which are assumed to be the

main cause of the brighter appearances with ageing. However, the patina of spruce timber with less oil gradually darkens with the increased building age. The experts in Huizhou-style ancient buildings name such phenomenon as 'patina of wood'. However, the surface reflectance of the 100-year spruce sample is lower than the 300-year sample, while the wavelength is under 700 nm, which does not coincide with the spectra of the cross-section, as Figure 7b,d shows, but the effect is minor.

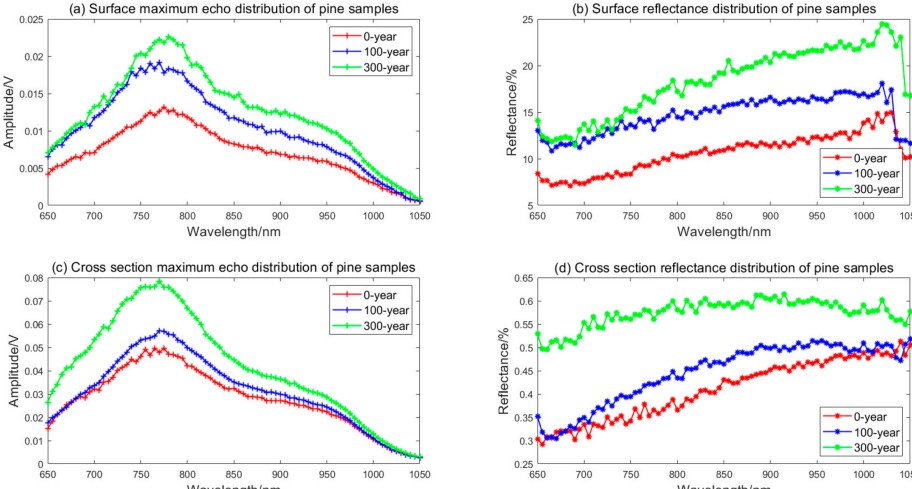

**Figure 6.** The comparison of pine samples surface and cross-section echo and reflectance. (**a**) Surface maximum echo distribution of pine samples. (**b**) Surface reflectance distribution of pine samples. (**c**) Cross section maximum echo distribution of pine samples. (**d**) Cross section reflectance distribution of pine samples.

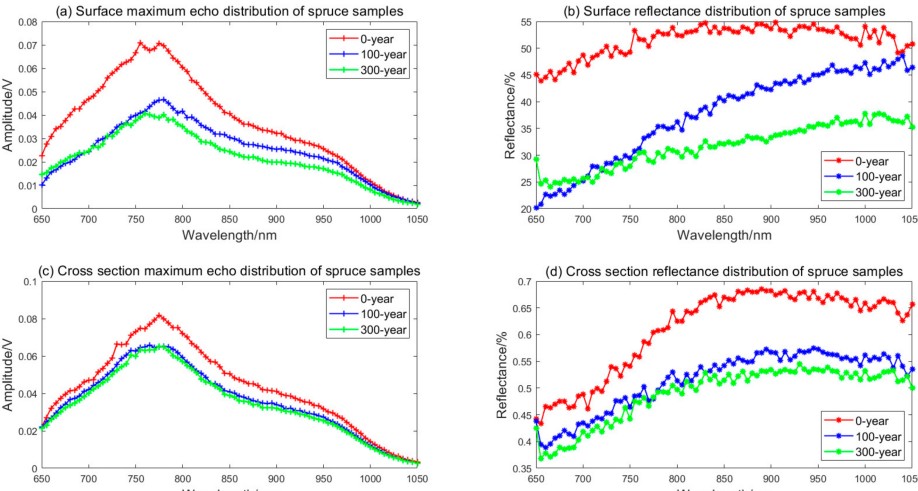

**Figure 7.** The comparison of spruce samples surface and cross-section echo and reflectance. (**a**) Surface maximum echo distribution of spruce samples. (**b**) Surface reflectance distribution of spruce samples. (**c**) Cross section maximum echo distribution of spruce samples. (**d**) Cross section reflectance distribution of spruce samples.

Further, we explored the feasibility of different wood species' surface spectra (pine, spruce, papyrifera, and hawthorn), which were taken from the same ancient building that was about 100 years old. Figure 8 shows the echo signal maximum and spectral reflectance of the 100-year samples, measured in the lab test, and it can be observed that different types of samples present unique surface echo maximum profiles, and their cross-section echo maximum profiles are closer than the surfaces, as Figure 8a,c shows. The reflectance distribution of the surface and cross-section in Figure 8b,d is

consistent with the echo signal maximum profiles. The surface spectral reflectance of the hawthorn sample is considerably higher than other samples, but its cross-section spectral reflectance values cross with curves of the spruce sample and papyrifera sample. The profile tendency in Figure 8b is consistent with Figure 8d; furthermore, the surface reflectance difference of the samples is more obvious than that of the cross-section. The conclusion can be drawn that the surface measurement of the timber component can be employed as a feature parameter to classify wood species directly.

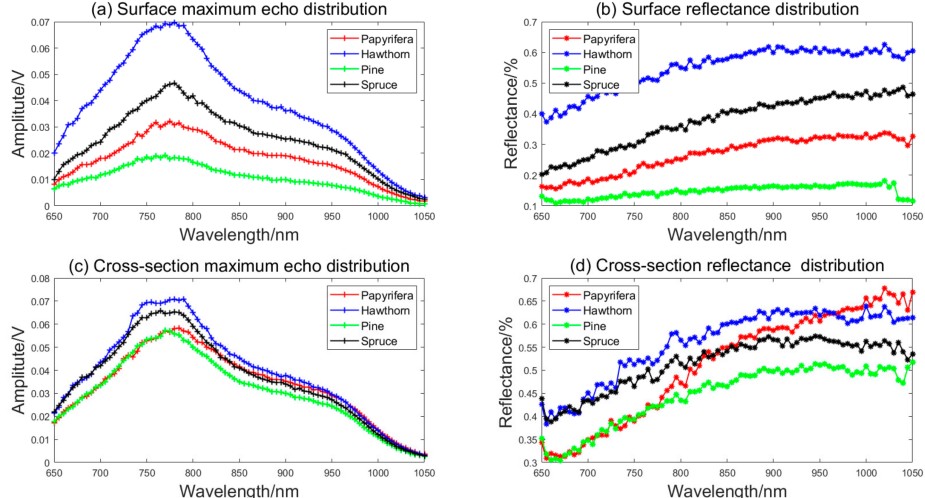

**Figure 8.** Echo and reflectance of different species of timber frame from the same 100-year old building. (**a**) Surface maximum echo distribution. (**b**) Surface reflectance distribution. (**c**) Cross section maximum echo distribution. (**d**) Cross section reflectance distribution.

### 4.2. Measurement of Aging Trends Based on HSL Reflectance

Figure 9 shows the reflectance difference between the surface and cross-section of the same timber component. The average deviations of pine samples (0-year, 100-year, and 300-year) are 30.2577, 29.7866, and 39.0866, respectively, and the corresponding average deviations of spruce samples are 9.6292, 13.886, and 17.2655, respectively. The reflectance difference between the surface and cross-section is increased with building-age as a whole. The ageing speed of the component surface is faster than cross-sections, and the reflectance is lower than the cross-sections.

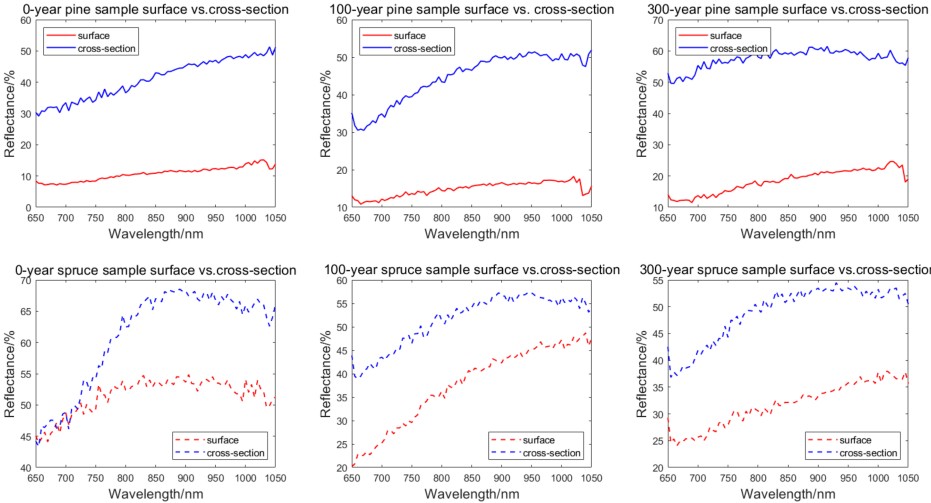

**Figure 9.** The reflectance comparison of surface and cross-section.

### 4.3. Preliminary Analyze Difference between Gnawed and Undamaged Samples

Seasonal cycles of dry and wet in the Huizhou area will facilitate fungal growth and insect attacks on timber structures, and the most common structural damage of Huizhou-style ancient buildings is gnawing by termites. It is important to find the damaged part and then replace it. The gnawed pine sample and the undamaged pine sample were cut off from the same architecture about 100 years. Figure 10 shows the echo signal maximum and reflectance difference between two samples; the surface reflectance and echo maximum of undamaged pine samples is larger than the gnawed samples, while the cross-section is adverse. As can be seen, the surface difference is larger than the cross-sections because the gnawed components degraded faster than the undamaged one.

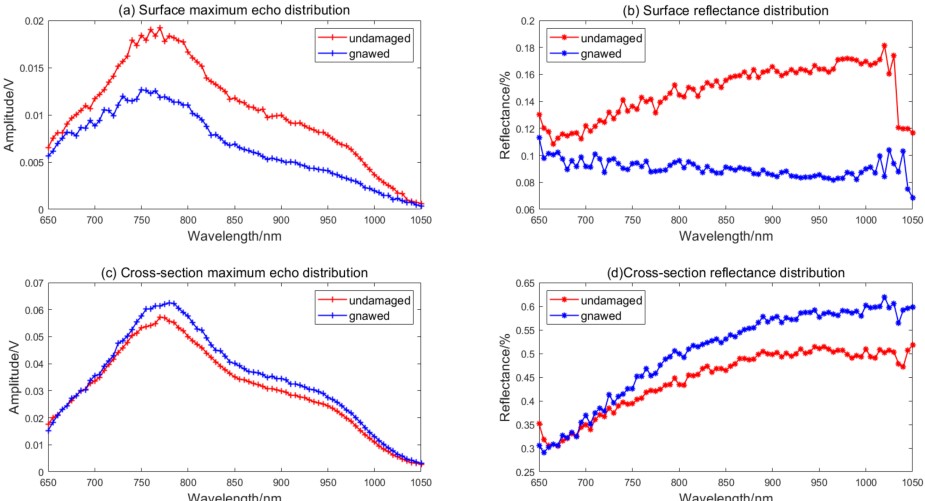

**Figure 10.** Comparison of the gnawed component with the undamaged one. (**a**) Surface maximum echo distribution. (**b**) Surface reflectance distribution. (**c**) Cross section maximum echo distribution. (**d**) Cross section reflectance distribution.

### 4.4. Classification Performance

In order to discuss the performance of timber component classification, six data sets, based on reflectance as a classification feature, were evaluated with multiple NB and SVM classifiers, they are:

- Dataset 1: pine samples including 0-year, 100-year, and 300-year.
- Dataset 2: spruce samples including 0-year, 100-year, and 300-year.
- Dataset 3: pine samples in Dataset 1 and gnawed sample from the 100-year building.
- Dataset 4: pine, spruce, papyrifera, and hawthorn samples taken from the same 100-year building.
- Dataset 5: Dataset 4 and the gnawed sample from the 100-year building.
- Dataset 6: all samples.

Table 3 shows the classification accuracy of six data sets with different classification methods, where the number in the table indicates the minimum number of spectral channels (MNSC) used to acquire 100% classification accuracy. The applied classification features in the table are the reflectance over the VIS to VNIR spectrum. In the testing phase, the channels are selected randomly, and the result is the average of 20 experiments.

The multiple NB classifiers can output satisfactory results; more than 15 channels of hyperspectral reflectance values are needed as the feature for classification operation, and classification accuracy reaches 100% in our data sets.

We anticipate the result that the multiple SVM classifiers can outperform with fewer spectral channels. The MNSC of Dataset 1 to require 100% classification accuracy is four with

SVM, and the MNSC of Dataset 2 is five. If we add the gnawed pine sample to the pine samples, such as in Dataset 3, the result is also excellent, where MNSC is five with SVM.

As can be seen, the same wood species samples with different building-ages can be classified easily, based on HSL measurements. If the timber components building-age can be classified, the original state and repaired state can be judged easily, which helps researchers to evaluate the integrity of the ancient structures.

**Table 3.** Classification results with reflectance.

| Dataset | MNSC (NB) | MNSC (SVM) |
|---------|-----------|------------|
| Dataset 1 | 24 | 4 |
| Dataset 2 | 15 | 5 |
| Dataset 3 | 25 | 5 |
| Dataset 4 | 28 | 8 |
| Dataset 5 | 27 | 7 |
| Dataset 6 | 29 | 9 |

Furthermore, we tested the classification results based on the reflectance of timber samples with the same building-age. The MNSC of Dataset 4 to require 100% classification accuracy is eight with the SVM classifier, and the MNSC of Dataset 5 is seven. The conclusion is drawn that we can classify different wood species based on HSL measurements in an ancient building. From the results of Dataset 3 and Dataset 5, we can discriminate the gnawed sample from other same-species undamaged samples; we can also classify the gnawed sample from different species in the same building. While the classification of all the samples we collected together had an MNSC of nine, we can achieve remarkable results of various timber components based on AOTF-HSL measurements, which provides the theoretical basis of distinguishing complicated states of a heritage building.

The classification results are excellent and we think four reasons might account for this: (1) 81-channel spectral information of AOTF-HSL provide abundant information for classification, (2) the small footprint size of AOTF-HSL affords exact measurements due to active measuring, (3) all samples are placed in the same distance under the controlled lab condition, and (4) the laser beams are projected perpendicularly on the surface of samples, in field testing, and the laser beam incident angle will affect the laser intensities.

From the results, we can observe that when several channels are employed, the 100% classification accuracy can be obtained. If the channels can be selected exactly, we can simplify the hardware. The high dimensional hyperspectral data also involve a high level of redundancy, implying inefficient data management and storage [45]. The identification and usage of a small number of data-dependent relevant bands would increase the applicability of the system. It will provide a basis in ancient building preservation.

Tables 4 and 5 depict the classification accuracy (%) of NB and SVM classifiers, based on the same datasets. It can be seen that the results vary with classifiers and datasets. Table 4 lists the classification accuracy (%) of six datasets with the NB classifier. We can observe that the classification accuracy increases with the number of channel increases. The different datasets have different results, and the classification result of dataset 2 is the best.

Table 5 illustrates the classification accuracy (%) of six datasets with the SVM classifier. We can conclude that the classification accuracy increases with the number of channels, and the hyperspectral measurements are beneficial for classification. SVM classification accuracy increases with the number of channels, especially in the case when the number of channels increases from two to three, the improvement is 11.06% on average. When the number of channels is beyond six, the accuracy is more than 90%. As mentioned before, Dataset 1/Dataset 2 has three classes, Dataset 3/Dataset 4 has four classes, Dataset 5 has five classes, and Dataset 6 has nine classes. As Tables 4 and 5 show, the average accuracy of different class numbers (3, 4, 5, and 9), based on SVM, is 94.6, 91.35, 89.13, and 87.21, respectively, and the average accuracy, based on NB, is 80.68, 75.55, 77.5, and 68.89, respectively; we

can draw a conclusion that the class number also influences the results of the classification and the accuracy decreases as the number of classes increase.

**Table 4.** Comparison NB classification accuracy for different datasets based on the number of channels.

| Dataset | NB Classification Accuracy (%) Based on Number of Channels | | | | | | |
|---|---|---|---|---|---|---|---|
| | **2** | **3** | **4** | **5** | **6** | **7** | **Mean** |
| **Dataset 1** | 59.81 | 63.32 | 75.05 | 75.1 | 79.09 | 81.07 | 72.24 |
| **Dataset 2** | 66.76 | 83.31 | 95 | 96.32 | 96.67 | 96.66 | 89.12 |
| **Dataset 3** | 58.24 | 65.06 | 71.66 | 80.04 | 80.16 | 81.34 | 72.75 |
| **Dataset 4** | 61.77 | 71.63 | 76.71 | 81.64 | 90.09 | 88.23 | 78.35 |
| **Dataset 5** | 58.27 | 71.67 | 74.97 | 85.01 | 86.67 | 88.38 | 77.5 |
| **Dataset 6** | 45.03 | 56.72 | 73.31 | 76.65 | 79.97 | 81.63 | 68.89 |
| **Mean** | 58.29 | 68.62 | 77.78 | 82.46 | 85.44 | 86.22 | |

**Table 5.** Comparison support vector machine (SVM) classification accuracy for different datasets based on the number of channels.

| Dataset | SVM Classification Accuracy (%) Based on Number of Channels | | | | | | |
|---|---|---|---|---|---|---|---|
| | **2** | **3** | **4** | **5** | **6** | **7** | **Mean** |
| **Dataset 1** | 71.78 | 90.05 | 100 | 100 | 100 | 100 | 93.64 |
| **Dataset 2** | 88.31 | 93.32 | 91.7 | 100 | 100 | 100 | 95.56 |
| **Dataset 3** | 85.03 | 93.26 | 95.06 | 100 | 100 | 100 | 95.56 |
| **Dataset 4** | 71.61 | 83.35 | 88.20 | 91.66 | 93.3 | 94.98 | 87.13 |
| **Dataset 5** | 73.41 | 84.98 | 91.66 | 91.54 | 93.21 | 100 | 89.13 |
| **Dataset 6** | 73.34 | 84.87 | 86.69 | 86.72 | 95.02 | 96.64 | 87.21 |
| **Mean** | 77.25 | 88.31 | 92.22 | 94.99 | 96.92 | 98.6 | |

## 5. Conclusions

The paper is designed an eye-safe 81-channel AOTF-HSL with 5 nm spectral resolution, covering from 650 to 1050 nm, and presented a feasibility study for ancient Huizhou-style architecture preservation with AOTF-HSL measurements. The hyperspectral spectra of ancient timber building components samples collected from two directions (surface and cross-section spectra). The feasibility of surface spectra for classification was demonstrated, based on the relationship between surface and cross-section spectra. Moreover, the multiple NB and SVM classifiers were performed with six different datasets, based on hyperspectral reflectance. The results further demonstrated the effectiveness of HSL in classifying building-age and wood species of ancient timber buildings—even the gnawed sample. Finally, we draw the following conclusions:

1. The tendencies of the surface and cross-section hyperspectral reflectance profiles are similar. The surface spectra can be used in the classification for the extraction of timber component physical properties.
2. Different wood species with the same building-age have different hyperspectral reflectance profiles; it is effective to classify wood species with AOTF-HSL measurements in order to provide a basis for further judging materials of ancient buildings.
3. The same wood species with different building-ages have different hyperspectral reflectance profiles; the classification accuracy of building-age reaches 100% by AOTF-HSL measurements, based on several channels, to provide the basis for further judging the original state and the repaired state of an ancient building.
4. The gnawed component classified from the undamaged components, based on the hyperspectral reflectance easily, will be a new, non-contact approach to detect the damaged component.

While some timber samples from several different ancient buildings are investigated in this research, the results can be considered suggestive. More samples in different conditions will be further

investigated for sounder research. The datasets are far from realistic in the complex real word for the efficient documentation of cultural heritage, but it is a preliminary result to guide the development of a specific system on spectra extraction. Our next research aim will be to extend the hyperspectral reflectance in 3D fine reconstruction, which can construct fine texture based on timber component building-age information and wood species classification.

**Author Contributions:** Conceptualization and writing original draft preparation: H.S. and Y.C.; methodology, supervision, project administration, funding acquisition, J.H., Y.C., Z.Y., J.C., S.W. and F.Y.; field test: W.L., H.W.; data analysis: C.J. and H.S.; writing—review and editing supported by E.P. All authors have read and agreed to the published version of the manuscript.

**Funding:** The author gratefully acknowledges the financial support from the Ministry of Science and Technology (2017YFC1405401), Academy of Finland projects "Centre of Excellence in Laser Scanning Research (CoE-LaSR) (307362)", "New laser and spectral field methods for in situ mining and raw material investigations (projects 292648, 314177 and 307929). Additionally, Chinese Academy of Science (181811KYSB20160113), the Chinese Ministry of Science and Technology (2015DFA70930), Shanghai Science and Technology Foundations (18590712600) and Beijing Municipal Science and Technology Commission (Z181100001018036) are acknowledged. This work was supported in part by Anhui Natural Science Research Foundation (KJ2019A0767, 1804d08020314), Research Program of Anhui Jianzhu University (JZ192007), Anhui and Jiangsu Province Key Laboratory Research Found (2017kfkt009, 2019-157), The research is also supported Research Council of Norway (287284).

**Acknowledgments:** We acknowledge the support of Anhui Huizhou Classical Garden Co., Ltd., (http://www.ahhzgj.com/) for the ancient timber component samples. And we thank Shunlai Yao (the expert in Huizhou-style ancient architecture) for advice.

**Conflicts of Interest:** The authors declare no conflict of interest.

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
