# Peer review of "Feasibility Study on Hyperspectral LiDAR for Ancient Huizhou-Style Architecture Preservation"

_remotesensing, doi:10.3390/rs12010088_

Round 1

Reviewer 1 Report

Half of the abstract can substantially be reduced.

Line 52: Add ‘(UNESCO)’

Lines 208 – 215 The calibration applied in Eq. (1) is too simple and tries to compare different wavelength units, which cannot be comparable. More rigorous spectral calibration procedures are expected to be implemented by the authors.

Please check statement on line 368. It seems to be the other way round. 

What about the LiDAR, with the XYZ coordinates? Where is the data acquisition dataset presented? No calibration about the XYZ coordinates is presented.

In general, too optimistic conclusions presented after several profiles with echo and reflections distributions. The authors should display graphically the classification results, mapping cultural heritage objects and features, not just some wavelength profiles of the objects/features. As it is presented, the results cannot be scientifically accepted. In addition, the authors omitted the most important part of a scientific paper, the discussion.

This paper cannot be presented in its present form.

Author Response

Point 1: Half of the abstract can substantially be reduced.

Response 1: We modified the abstract, but we cant reduce it to half. The major reason the article is a more multidisciplinary research, we hope to present the HSL application in Huizhou style architecture Preservation clearly.

Point 2: Line 52: Add ‘(UNESCO)’

Response 2: Fixed, thank you for suggestion.

Point 3: Lines 208 – 215 The calibration applied in Eq. (1) is too simple and tries to compare different wavelength units, which cannot be comparable. More rigorous spectral calibration procedures are expected to be implemented by the authors.

Response 3: We have already discuss the calibration for the previous study [Reference 28: Hakala, T., Suomalainen, J., Kaasalainen, S., & Chen, Y.W. Full waveform hyperspectral LiDAR for terrestrial laser scanning. Optics express, 2012 20(7), 7119-7127; Reference 30: Chen, Y.W.; Li, W.; Hyyppä, J.; Wang, N.; Jiang, C.H.; Meng, F.R.; Tang, L.L.; Puttonen, E.; Li, C.R. A 10-nm Spectral Resolution Hyperspectral LiDAR System Based on an Acousto-Optic Tunable Filter. Sensors 2019, 19,1620]. It is an effective solution if the target distance is fixed and laser beam project perpendicularly on the surface of the target in short distance. In the paper, before collecting the reflectance spectra of timber samples, the white reference panel with a reflectance of 70% placed as the identical distance from the target as the samples away from the HSL system was captured first, and then the capture would be again performed to weaken the influence of some external factors and we selected the laser beams were projected perpendicularly on the surface of samples.

Point 4: Please check statement on line 368. It seems to be the other way round.

Response 4: The statement about Figure 11 (updated to Table 3) is on line 366( updated to line 387), and we added the relation of the class number and accuracy in the revise version. We analyzed how the number of classes influence the classification results, as the dataset 1 and dataset 2 had 4 classes samples,and dataset 3 and dataset 4 had 4 classes samples, and dataset 5 had 5 classes samples, and dataset 5 had 9 classes samples. The classification accuracy based on SVM is 94.6, 91.35, 89.13, 87.21 respectively, and the accuracy based on NB is 80.68, 75.55, 77.5, 68.89 respectively, as list in Table 3. So we drew a conclusion that the class number also influence the results of the classification, the accuracy decrease with the number of classes increase.

Point 5: What about the LiDAR, with the XYZ coordinates? Where is the data acquisition dataset presented? No calibration about the XYZ coordinates is presented.

Response 5: The range accuracy also has been discussed in the paper [Reference 21( updated to Reference 30):Chen, Y.W.; Li, W.; Hyyppä, J.; Wang, N.; Jiang, C.H.; Meng, F.R.; Tang, L.L.; Puttonen, E.; Li, C.R. A 10-nm Spectral Resolution Hyperspectral LiDAR System Based on an Acousto-Optic Tunable Filter. Sensors 2019, 19,1620]. There is very minor range difference between different spectral channels. With current sampling speed of employed oscilloscope, the range resolution is 7.5 mm, however, due to current setup, the exact optical length for the system is impossible to be measured. Currently the system is installed on a rotated platform with angular step of better than 0.1 degrees, we can obtain 3D spatial information. Using traditional laser scanner operating in terrestrial laser scanning mode, very precise 3D model can be built with millimeter accuracy in spatial domain. However, such detailed and precise model cannot fulfill the following two mission. 1) classifying the wood species based on spatial information and intensity information 2) cannot assess the building age from intensity information from a single-wavelength light source, not to mention how to detect gnawed woods. So all current commercial LiDAR is not application for such type of studies. Actually the HSL is designed for forest related research, for example, for pest control. We found that the developed HSL can detect the pest damaged wood. Which give us the hint whether it can be used for examining wood from Huizhou Style architecture,and and our next research aim will be extending the hyperspectral reflectance in 3D fine reconstruction.

Point 6: In general, too optimistic conclusions presented after several profiles with echo and reflections distributions. The authors should display graphically the classification results, mapping cultural heritage objects and features, not just some wavelength profiles of the objects/features. As it is presented, the results cannot be scientifically accepted. In addition, the authors omitted the most important part of a scientific paper, the discussion.

Response 6: Of course, we prefer to test the newly developed system into a real ancient Huizhou-style architecture, however, as the hardware presented in following figure presents. The mobility of system should be necessarily improved (the prototyped system is in Beijing which is 2000 km far from the site where most ancient Huizhou style architecture locate). As we know with traditional terrestrial Laser scanning, better than a few millimeters accuracy model of architecture can be easily achieve. Thus in this study, we do not focus on the spatial resolution and calibration in spatial domain.

And in this feasibility research, we got these ancient wood samples taken from the ancient architecture (due to the renovation and replacement). We agreed that it might be too optimistic conclusions with these test samples. However, we believe that as the first publication in HSL on ancient architecture preservation. This paper might open a door and offer a novel potential solution for historical wood architecture preservation. So we still believe that this publication has its unique reference value for other researchers.

Reviewer 2 Report

The paper entitled as "Feasibility Study on Hyperspectral LiDAR for Ancient Huizhou Style Architecture Preservation" presents an interesting topic dealing with the classification of wooden material based on hyperspectral ground measurements. While I am in favour of seeing this paper being published I think the authors should address some of the points raised below so as to improve the overall quality of their work:

1) Please add an introduction section which related state of the art is provided to the readers. Several studies can be found dealing with the classification of single and multi beam terrestrial laser scanners, which i think it is appropriate to be presented in the begining of the paper. The current section 1 should move next as the case study

2). Please elaborate more the difficulties you faced for you work i.e. why other methods are not appropriate? while this is presented somehow in lines 104-118 i think more details are needed. Please see also my previous comment, which can be linked together so as to higlight you work

3) Regarding the system that the authors have built: please provide in a a table the major characteristics of your system; eg. spectral range; spectral resolution; FOV; scan time etc. Some of these characteristics are scattered in the text and difficult to sum up with the general concept of the instrument.

4). Why building up a new system? I think this should be highlighted since this adds more on the novelty of your work. Are existing instruments too expensive, not applicable for such types of studies?

5) Figure 11 and Figure 12. Please provide the confusion matrix table and not the graphs. Indicate the overall accuracy of your measurements. 

6) Discussion section is needed before going to a more general closing remarks section (Conclusion).

7) Expand you references (see also my point 1).

Overall i think the authors have done a great work which can be improved and presented to the readers

Author Response

Point 1: Please add an introduction section which related state of the art is provided to the readers. Several studies can be found dealing with the classification of single and multibeam terrestrial laser scanners, which i think it is appropriate to be presented in the begining of the paper. The current section 1 should move next as the case study.

Response 1: We added some research which using laser scanning for heritage protection into the revised version. Terrestrial laser scanners provide high spatial resolution point data with intensity information on selected wavelength, and are excellent instruments for monitoring deform and 3D construction, however,they are inadequate for research of fine texture and patina of wood, wood species and temporal evolution evaluation.

Point 2: Please elaborate more the difficulties you faced for you work i.e. why other methods are not appropriate? while this is presented somehow in lines 104-118 i think more details are needed. Please see also my previous comment, which can be linked together so as to higlight you work.

Response 2: Fixed, we presented the other methods (lines 104-118) in more details.

Point 3: Regarding the system that the authors have built: please provide in a a table the major characteristics of your system; eg. spectral range; spectral resolution; FOV; scan time etc. Some of these characteristics are scattered in the text and difficult to sum up with the general concept of the instrument.

Response 3: We added major parameter specifications of our system, as Table 1 in the revised version.

Point 4: Why building up a new system? I think this should be highlighted since this adds more on the novelty of your work. Are existing instruments too expensive, not applicable for such types of studies?

Response 4: There is not existing commercial hyperspectral or multispectral LIDAR for such purpose. Using traditional laser scanner operating in terrestrial laser scanning mode, very precise 3D model can be built with millimeter accuracy in spatial domain. However, such detailed and precise model cannot fulfill the following two mission. 1) classifying the wood species based on spatial information and intensity information 2) cannot assess the building age from intensity information from a single-wavelength light source, not to mention how to detect gnawed woods. So all current commercial LiDAR is not application for such type of studies. Actually the HSL is designed for forest related research, for example, for pest control. We found that the developed HSL can detect the pest damaged wood. Which give us the hint whether it can be used for examining gnawed wood from Huizhou Style architecture.

Point 5: Figure 11 and Figure 12. Please provide the confusion matrix table and not the graphs. Indicate the overall accuracy of your measurements.

Response 5: Fixed, Thank you for suggestion, we updated figure to the confusion matrix table, and we also indicate the overall accuracy of measurements.

Point 6: Discussion section is needed before going to a more general closing remarks section (Conclusion).

Response 6: Fixed, Thank you. Our discussion section was included in the Experiments and analysis section.

Point 7: Expand you references (see also my point 1).

Response 7: Fixed.Thank you.

Reviewer 3 Report

GENERAL COMMENT

The industry and the research community urgently need a technique for rapidly and accurately classifying the wood material and discriminating their building-age. This paper focuses on the usability of 81-channel hyperspectral LiDAR (HSL) to tackle the issues. Based on the HSL measurements Authors analysed the relationship between the surface and cross-section spectral profiles of timber components from different ancient architecture built in early Qing dynasty (~300 years), late Qing dynasty (~100 years) 35 and nowadays. Authors described the feasibility of using surface spectra of timber components for classification purpose. The topic is of current interest and the testing reported could produce valuable outcomes, anyway the research presents the following issues:

Keywords

The keyword “Original state” is not necessary in this paper. Authors can skip those words.

Introduction

Authors correctly presented state of the art. Information given on the Fig. 1 and Fig. 2 are basic and not necessary in a scientific paper. Authors should add the papers of other scientists concerning with the assessment of the wood age.

Samples and Methods

Authors should add more information about properties of the wood species (pine, spruce, papyrifera and Hawthorn wood) used in experiments. How many samples were used for the investigations?

Conclusions

First sentence “…The paper investigated an eye-safe AOTF-HSL with 5nm spectral resolution, covering from 650 nm to 1050 nm, and 81-channel spectral information was acquired together. We presented a feasibility study for ancient Huizhou-style architectures preservation by multiple NB and SVM classifiers with AOTF-HSL spectra. We collected AOTF-HSL measurements of different samples from different building-age ancient buildings. Firstly, the feasibility of surface spectra was demonstrated based on the relationship between the surface and cross-section spectra. Moreover, the multiple NB and SVM classifiers were performed with six different datasets based on hyperspectral reflectance. The results further demonstrated the effectiveness of the HSL in classifying building-age and wood species of ancient timber frames, even the gnawed sample. Finally, we draw the following conclusions: the modelling library developed in this work is sufficiently generic to accommodate modelling of different commercial digesters. This opens up the opportunities for pulp mills to implement MPC for problematic pulp digester control…” should be removed from the conclusions chapter and placed in the chapter 4.

Paper can be published after minor changes and additions.

Author Response

Point 1: Keywords:The keyword “Original state” is not necessary in this paper. Authors can skip those words.

Response 1: Fixed. Thank you for suggestion.

Point 2: Introduction:Authors correctly presented state of the art. Information given on the Fig. 1 and Fig. 2 are basic and not necessary in a scientific paper. Authors should add the papers of other scientists concerning with the assessment of the wood age.

Response 2: The major reason we put these two figures is that 1) it is a more multidisciplinary research, some researchers with different background might also interested on the method and have no idea about Huizhou style architecture. 2) explain also why pine, spruce, hawthorn, and papyrifera are selected for classification, because they are most common tree species suitable for construction in ancient time 3) it also explained why gnawing happens for example, the wood building in north part of China has less gnawed problem due to drier weather, some temples built more than 1000 years ago still keep its original shape. Such location based problem is due to the season and climate ,which shown in the Figure 1 and Figure 2. 2) We traced the papers of other scientists about the assessment of the wood age, which can evaluate the wood or tree age exactly, but our concerning is different, and our next research aim will be extending the hyperspectral reflectance in 3D fine reconstruction.

Point 3: Samples and Methods: Authors should add more information about properties of the wood species (pine, spruce, papyrifera and Hawthorn wood) used in experiments. How many samples were used for the investigations?

Response 3: We selected nine samples with four species wood in the investigations, and we got these ancient wood samples taken from the ancient architecture (due to the renovation and replacement) in this feasibility research, we will further explore more samples.

Point 4: Conclusions: First sentence “…The paper investigated an eye-safe AOTF-HSL with 5nm spectral resolution, covering from 650 nm to 1050 nm, and 81-channel spectral information was acquired together. We presented a feasibility study for ancient Huizhou-style architectures preservation by multiple NB and SVM classifiers with AOTF-HSL spectra. We collected AOTF-HSL measurements of different samples from different building-age ancient buildings. Firstly, the feasibility of surface spectra was demonstrated based on the relationship between the surface and cross-section spectra. Moreover, the multiple NB and SVM classifiers were performed with six different datasets based on hyperspectral reflectance. The results further demonstrated the effectiveness of the HSL in classifying building-age and wood species of ancient timber frames, even the gnawed sample. …” should be removed from the conclusions chapter and placed in the chapter 4.

Response 4: Followed the reviewer’s comment and make a lighter conclusion on the chapter 5.

Round 2

Reviewer 1 Report

Dear authors, it's a pity you didn't show explicitly the benefits of your new device. Besides, the dataset is far from realistic in the complex real word for the efficient documentation of cultural heritage. Therefore, I'm afraid that the present manuscript is not ready to be published in this high impact journal.

Please try to follow strictly the recommendations suggested by the reviewers.

Author Response

The data sets are far from realistic in the complex real word for the efficient documentation of cultural heritage, but it is a preliminary result to guide the development of a specific system on spectra extraction. In the paper, we didn’t discuss feature extraction based on geometry, we just discussed features based on the backscattered reflectance spectra, while the backscattered reflectance are combined with the corresponding time-of-flight and concurrent scanner orientation, a hyperspectral point cloud is produced. Many researchers built accurate 3D models with 1 cm precision easily, employing TLS single or multiple spectral scanner, so 3D modeling ability was not repeated in the paper. Our next research aim will be extending the hyperspectral reflectance in 3D fine reconstruction, which can construct fine texture based on timber component building-age information and wood species classification.

We counted and implemented the comments of reviewers in detail.

Reviewer 2 Report

The authors have addressed my concerns raised in the 1st round. I think the paper has been improved presenting all the details for the specific study.

Author Response

Thank you .We revised manuscript meticulously, and we hope to present our study in detail.